# NF-YA transcription factors suppress jasmonic acid-mediated antiviral defense and facilitate viral infection in rice

**Xiaoxiang Tan[1,2]‡, Hehong Zhang[2]‡, Zihang Yang[2], Zhongyan Wei[2], Yanjun Li[2], Jianping Chen[1,2]\*, Zongtao Sun[2]\***

**1** College of Plant Protection, Northwest A&F University, Yangling, Shaanxi, China, **2** State Key Laboratory for Managing Biotic and Chemical Threats to the Quality and Safety of Agro-products, Institute of Plant Virology, Ningbo University, Ningbo, China

‡ These authors contributed equally to this work.
\* jianpingchen@nbu.edu.cn (JC); sunzongtao@nbu.edu.cn (ZS)

## Abstract

NF-Y transcription factors are known to play many diverse roles in the development and physiological responses of plants but little is known about their role in plant defense. Here, we demonstrate the negative roles of rice *NF-YA* family genes in antiviral defense against two different plant viruses, *Rice stripe virus* (RSV, *Tenuivirus*) and *Southern rice black-streaked dwarf virus* (SRBSDV, *Fijivirus*). RSV and SRBSDV both induced the expression of *OsNF-YA* family genes. Overexpression of *OsNF-YAs* enhanced rice susceptibility to virus infection, while *OsNF-YAs* RNAi mutants were more resistant. Transcriptome sequencing showed that the expression of jasmonic acid (JA)-related genes was significantly decreased in plants overexpressing *OsNF-YA* when they were infected by viruses. qRT-PCR and JA sensitivity assays confirmed that OsNF-YAs play negative roles in regulating the JA pathway. Further experiments showed that OsNF-YAs physically interact with JA signaling transcription factors OsMYC2/3 and interfere with JA signaling by dissociating the OsMYC2/3-OsMED25 complex, which inhibits the transcriptional activation activity of OsMYC2/3. Together, our results reveal that OsNF-YAs broadly inhibit plant antiviral defense by repressing JA signaling pathways, and provide new insight into how OsNF-YAs are directly associated with the JA pathway.

## Author summary

In the arms race between host and viruses, plants have sophisticated defense strategies to cope with pathogenic invasion. Our previous research has shown that plant hormones, especially JA signaling, play essential roles in plant antiviral defense. Therefore, identification of the regulatory components and network of JA signaling is an important contribution to understanding plant antiviral defense. Here, we found that NF-YAs widely repress plant resistance against different types of viruses. We show that NF-YAs inhibit JA-mediated antiviral defense by dissociating the OsMYC2/3-OsMED25 complex, and inhibiting

**Data Availability Statement:** Transcriptome datasets have been submitted to NCBI under the BioProjects (PRJNA790476). All other relevant data

are within the manuscript and its Supporting information files.

**Funding:** This work was supported by the National Key Research and Development Plan of China to ZS (2021YFD1400500), the Natural Science Foundation of China to ZS (32022072), YL (32172416) and ZW (32100103), Zhejiang Provincial Natural Science Foundation to ZS (LZ22C140001), and the K. C. Wong Magna Fund in Ningbo University to ZS. The funders had no role in the study design, data collection and analysis, decision to publish, or preparation of the manuscript.

**Competing interests:** The authors have declared that no competing interests exist.

the transcriptional activation activity of OsMYC2/3. Together, our results greatly expand our knowledge about the function of NF-YA family genes associated with JA-mediated antiviral immunity.

## Introduction

Among the various viruses that cause severe economic losses in rice production around the world, *Southern rice black-streaked dwarf virus* (SRBSDV) and *Rice stripe virus* (RSV) are currently widespread in east Asia and relatively well-studied. SRBSDV belongs to the genus *Fijivirus* in the family *Reoviridae* and is mainly transmitted in a persistent propagative manner by *Sogatella furcifera* (the white-backed planthopper, WBPH) [1,2]. The SRBSDV genome has 10 segments (S1–S10) of linear double-stranded RNA (dsRNA). Plants infected by SRBSDV are severely stunted and have excessive tillering. RSV is classified in the genus *Tenuivirus* (family *Phenuiviridae*) and is transmitted in a persistent propagative manner by *Laodelphax striatellus* (the small brown planthopper, SBPH) [3,4]. RSV has a single-stranded RNA (ssRNA) genome of four segments. RSV infection usually causes chlorotic or necrotic stripes on newly developed leaves, which is followed by wilting and stunting.

Plant hormone signaling pathways play important roles in response to infection of rice by viruses. For example, rice *STV11* encodes a sulfotransferase, which can catalyze the conversion of salicylic acid (SA) into the more effective sulfonated SA (SSA) and enhance the resistance of rice to RSV [5]. *Rice dwarf virus* (RDV)-encoded Pns11 protein promotes ethylene production which weakens the plant resistance to viral infection [6]. The multiple ARF transcription factors in rice have diverse regulatory functions on rice antiviral defense [7]. Our previous studies showed that different phytohormones play diverse roles in antiviral defense response. For example, the jasmonic acid (JA) pathway positively regulated rice defense, and suppressed brassinosteroids (BR)-mediated susceptibility to viral infection [8]. Abscisic acid (ABA) increased the susceptibility of rice to virus by inhibiting JA pathways [9]. BR mediates the susceptibility to *Rice black-streaked dwarf virus* (RBSDV) infection by inhibiting the plant defense response mediated by JA and SA and reducing the accumulation of reactive oxygen species (ROS) mediated by peroxidase [10]. Activation of the JA pathway was part of the auxin signalling-mediated antiviral defense [11]. It has recently been shown that JA signaling activates RNA silencing, and synergistically promotes rice antiviral defense [12]. The accumulation of miR319 by *Rice ragged stunt virus* (RRSV) in rice suppresses JA-mediated defense to promote viral infection and symptom development [13]. These studies all illustrate that JA signaling plays essential roles in rice antiviral defense. Therefore, identification of the regulatory components and network of JA signaling is an important contribution to understanding plant antiviral defense.

Nuclear factor Y (NF-Y) transcription factors act as essential regulators of growth developmental process and stress responses. Because they recognize the CCAAT-box, NF-Ys are also called CCAAT-binding factors (CBFs) or Heme Activator Proteins (HAPs) [14–16]. Classically, NF-Y is a heterotrimeric transcription factor complex consisting of three subunits: NF-YA (HAP2/CBF-B), NF-YB (HAP3/CBF-A) and NF-YC (HAP5/CBF-C) subunits, which are highly conserved in most species [17,18]. Usually, NF-YB and NF-YC dimerize in the cytoplasm and move to interact with NF-YA in the nucleus [19]. This hetero-trimeric complex then binds to the CCAAT-box element to activate or repress gene expression. NF-Y transcription factors are encoded by a single-gene in animals and yeast. However, in plants, each NF-Y subunit type is encoded by 10 or more genes due to gene duplication events [20]. For example, the *Arabidopsis* genome encodes 10 *NF-YA*, 13 *NF-YB* and 13 *NF-YC* genes [21], while rice has

11 *NF-YA*, 11 *NF-YB* and 12 *NF-YC* genes [22]. This multiplicity suggests a substantial potential regulatory capacity.

NF-Y family members are characterized by the presence of highly conserved domains. NF-YA proteins have two conserved domains (subunit interactions and DNA binding domains). The subunit interactions domain is crucial for the interaction with NF-YB and NF-YC, and the DNA binding domain is required for sequence-specific binding to CCAAT-boxes. Numerous studies have identified the *NF-Y* genes that play vital roles in plant developmental and physiological responses. The first plant NF-Y regulating development was identified and reported in *Arabidopsis* [23] and was found to be involved in embryogenesis [24]. In *Arabidopsis*, NF-YC3, NF-YC4 and NF-YC9 can interact with CONSTANS (CO), a key gene regulating flowering, to promote the photoperiod-dependent flowering process [25]. The NF-Y complex promotes flavonoid biosynthesis by binding CCAAT elements and recruiting histone methyltransferase or demethylase to regulate H3K27me3 levels in tomato [26]. In rice, OsNF-YB1 forms a heterotrimeric complex with OsNF-YC12 and bHLH144 to directly regulate the transcription of *Wx* and regulate the synthesis of starch [27]. OsNF-YA7 was induced by drought and improved rice tolerance to drought stress [28]. In addition, some NF-YA family members were shown to modulate plant defense against pathogens. For example, overexpression of *OsHAP2E* (*OsNF-YA2*) conferred resistance in rice to the fungal pathogen *Magnaporthe oryzae* and the leaf blight bacterium *Xanthomonas oryzae* pv. *Oryzae* [29]. Down-regulation of the OsNF-YA family members by overexpressing miR169a makes rice hypersusceptible to *M. oryzae* [30]. These studies showed that OsNF-YAs positively regulate plant defense against fungal and bacterial pathogens but the relationship between NF-YAs and viral pathogens has never been investigated, and it is unclear whether and how NF-YAs affect JA-mediated antiviral defense.

In this study, we explored the roles of OsNF-YAs in defense against viruses. In contrast to their positive roles against fungal and bacterial pathogens, OsNF-YAs negatively regulated rice antiviral defense against two different RNA viruses, RSV and SRBSDV. We found that OsNF-YAs directly interacted with OsMYC2/3, which was shown to be the key transcription factor in the JA pathway. We provide evidence that OsNF-YAs destroy the interaction between OsMYCs-OsMED25 complexes, and repress the transcriptional activation activity of OsMYC2/3, finally impairing JA-mediated antiviral defense.

## Results

### Infection by RSV or SRBSDV induces the expression of *OsNF-YAs*

To investigate the roles of *OsNF-YA* in defense against viruses, the expression levels of *OsNF-YA* family genes were analyzed following infection by viruses. *Oryza*. *sativa* L. *japonica* Nipponbare (NIP) was used for viral inoculation experiments because it is susceptible to both RSV and SRBSDV. Results of qRT-PCR showed that a number of *OsNF-YA* family genes, including *OsNF-YA1*, *OsNF-YA2* and *OsNF-YA10*, were significantly induced 30 day after inoculation with RSV, a single-stranded RNA virus (Fig 1A). Similarly, the expression level of most *OsNF-YAs* was obviously increased 30 dpi with SRBSDV, a double-stranded RNA virus (Fig 1B). The finding that two different viruses both induced the expression of *OsNF-YA* family genes, suggested that *OsNF-YAs* might be generally involved in the interaction between rice and viruses.

### Overexpression of *OsNF-YAs* increased susceptibility to infection by RSV or SRBSDV

We next constructed transgenic rice lines overexpressing *OsNF-YAs* and inoculated them with viruses. Stable transgenic lines expressing three representative *OsNF-YA* genes (*OsNF-YA1*,

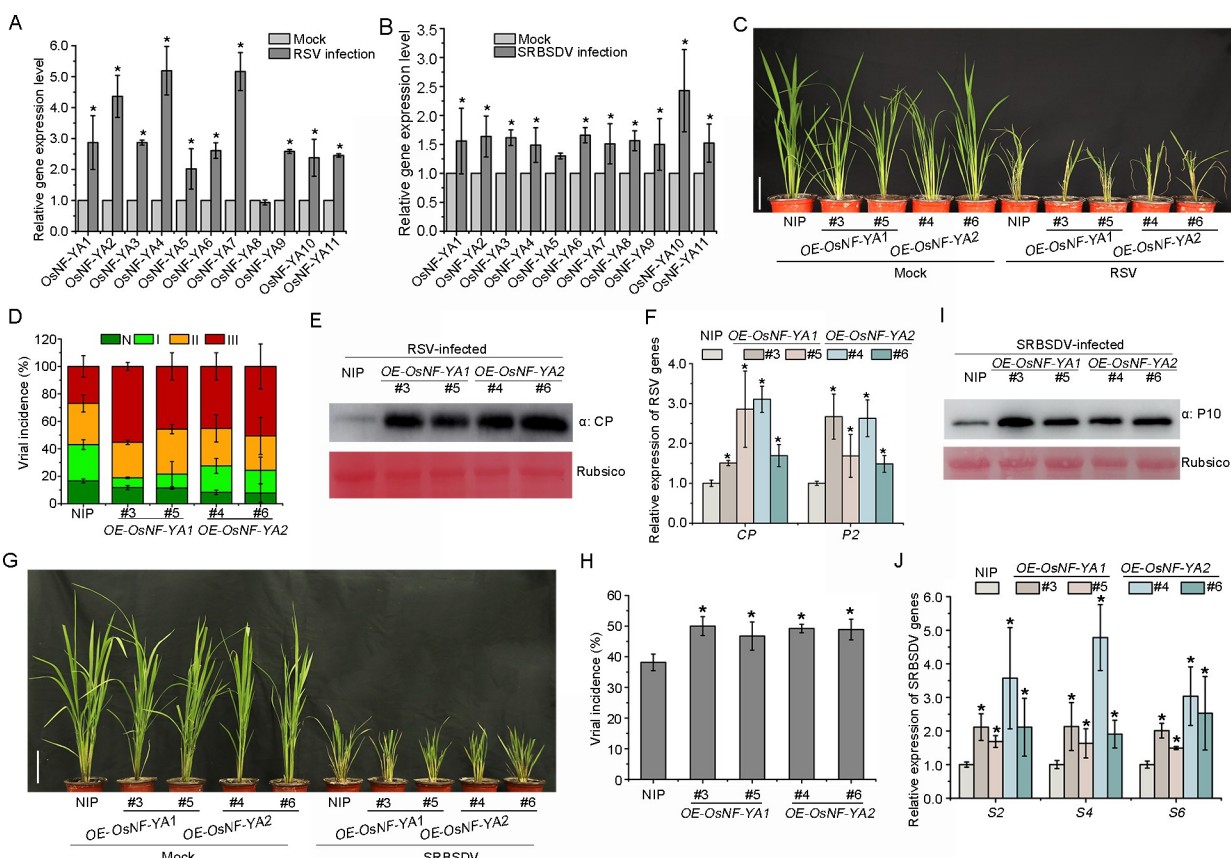

**Fig 1. OsNF-YAs negatively modulate rice resistance to RSV and SRBSDV.** A,B. The relative expression levels of *OsNF-YA* family genes after infection by (A) Rice stripe virus (RSV) and (B) Southern rice black-streaked dwarf virus (SRBSDV). RNA was extracted from infected plants at 30 days post inoculation (dpi). Data are shown as relative expression level of *OsNF-YA* family genes in infected plants in comparison with mock plants. C. Symptoms of RSV infection in *OE-NF-YAs* transgenic lines compared with NIP. The phenotypes were observed and photos were taken at 20 dpi. (Scale bars, 10 cm.) D. The percentage of RSV-inoculated plants with different degrees of disease. N, Healthy; I, mild mosaic; II, severe mosaic; III, wilting. E. The accumulation of RSV CP protein in RSV-infected plants determined by western blotting. Rubsico was used as an internal reference. F. qRT-PCR results showing the relative expression levels of RSV RNAs (*CP* and *P2*) in virus-infected plants. G. Symptoms of SRBSDV infection in *OE-NF-YAs* transgenic lines. The phenotypes were observed and photos were taken at 30 dpi. (Scale bars, 10 cm.) H. Incidence of SRBSDV in inoculated *OE-NF-YAs* transgenic lines compared with NIP. I. The accumulation of SRBSDV P10 protein in virus-infected plants determined by western blotting. Rubsico was used as an internal reference. J. qRT-PCR results showing the relative expression levels of SRBSDV RNA segments (*S2*, *S4*, *S6*) in virus-infected plants. *OsUBQ5* was used as the internal reference gene to normalize the relative expression. Values shown are the means ± SD of 3 biological replicates. Significant differences were identified using Fisher's least significant difference tests. *At the top of columns indicates significant difference at $p \leq 0.05$.

*OsNF-YA2*, *OsNF-YA10*) were created in the NIP background and the expression levels of these genes were verified by qRT-PCR (S1A Fig). The homozygous T2 generation transgenic rice lines *OE-OsNF-YA1#3* and *#5*, *OE-OsNF-YA2#4* and *#6*, *OE-OsNF-YA10#1* and *#2* were then first inoculated with RSV. The severity of RSV symptoms that developed on each plant was assessed using the four grades: healthy (N), mild mosaic (I), severe mosaic (II) or wilting (III) (S2 Fig). Symptoms were more severe in the overexpression lines than in the NIP control (Figs 1C and S3A). The transgenic lines had severe dwarfing, and many plants were wilting (grade III) or had severe mosaic (grade II) (Figs 1D and S3B). Similarly, the accumulation of RSV *RNAs* (*CP* and *P2*) and CP protein in the overexpressed transgenic lines were significantly higher than those of NIP (Figs 1E and 1F, and S3C and S3D).

The *OE-OsNF-YAs* transgenic lines were then inoculated with the double-stranded RNA virus SRBSDV. More severe stunting symptoms were observed in SRBSDV-infected transgenic

lines than in control plants (Fig 1G). Viral incidence was significantly higher in both *OE-OsNF-YA1* and *OE-OsNF-YA2* transgenic lines than in NIP plants (Fig 1H). qRT-PCR and western blot experiments showed that viral RNA transcripts (*S2*, *S4* and *S6*) and caspid protein P10 were all obviously more abundant in the transgenic lines than in the controls (Fig 1I and 1J). Thus, overexpression of *OsNF-YAs* made rice more susceptible to both the ssRNA virus RSV and the dsRNA virus SRBSDV.

To investigate any effect of the *OsNF-YAs* transgenic plants on the insect vectors, we tested the survival of SPBH and WBPH on the different plants over a 5-day period but found no significant differences in survival between the transgenic and control plants (S1C and S1D Fig).

## Suppressing the expression of *OsNF-YAs* increased resistance to viruses

To further study the relationship between OsNF-YAs and viral infection, we next constructed stable transgenic rice lines with decreased expression of *OsNF-YAs* by RNA interference. Because of the functional redundancy among *OsNF-YAs*, we constructed these lines using a conserved region of the *OsNF-YA* family genes and obtained lines named *RNAi-OsNF-YA#7* and *RNAi-OsNF-YA#8* that had efficiently suppressed expression of *OsNF-YAs* (S1B Fig). When these lines were challenged with RSV, symptoms were less severe than in the NIP controls, with less stunting, and a larger percentage of plants that were healthy (grade N) or which had only slight mosaic symptoms (grade I) (Fig 2A and 2B). Consistently, RSV RNAs (*CP* and *P2*) and CP protein were significantly lower in *RNAi-OsNF-YA* lines than in wild type NIP (Fig 2C and 2D). The *RNAi-OsNF-YA* lines were also more resistant to SRBSDV, with less dwarfism (Fig 2E) and significantly lower viral incidence (Fig 2F) than in the control NIP (Fig 2F). qRT-PCR and western blot experiments also confirmed that the accumulation of SRBSDV RNA transcripts (*S2*, *S4* and *S6*) and CP protein were significantly reduced in the transgenic lines (Fig 2G and 2H). Thus repression of the expression of *OsNF-YAs* increased rice resistance to both RSV and SRBSDV.

## Transcriptomic analysis suggests that OsNF-YAs modulate the JA pathway

To understand how OsNF-YAs negatively regulate the antiviral immune response, we performed a genome-wide transcriptomic analysis on NIP and *OE-OsNF-YA2#4* transgenic plants infected with RSV at 20 dpi. In order to reduce the likelihood of false positives, we used a *p*-value less than 0.05 as the cutoff criterion. We identified 4620, 971 and 7013 differentially up-regulated genes in the respective subsets *OE-OsNF-YA2*-CK versus NIP-CK, *OE-OsNF-YA2*-RSV versus NIP-RSV, and NIP-RSV versus NIP-CK (Fig 3A). Venn diagram analysis showed that about half of the genes (429/971) were specifically up-regulated in *OsNF-YA2* transgenic plants after RSV infection (Fig 3A). Hierarchical clustering analysis showed these 429 genes were indeed specially activated in the *OsNF-YA2* plants compared to the control NIP plants (Fig 3B). Gene ontology (GO) analysis clearly revealed that these genes are highly enriched in the categories of plasma membrane, chloroplast, DNA-binding transcript factor and protein phosphorylation (Fig 3C). Meanwhile, we identified 1483, 754 and 2753 differentially down-regulated genes in the respective subsets *OE-OsNF-YA2*-CK versus NIP-CK, *OE-OsNF-YA2*-RSV versus NIP-RSV, and NIP-RSV versus NIP-CK. Interestingly, almost all the down-regulated genes (634/754) were manipulated by the OsNF-YA2 in the context of RSV infection (Fig 3D). Hierarchical clustering revealed that these genes were obviously suppressed in the *OE-OsNF-YA2* plants compared to the control NIP plants under RSV infection (Fig 3E). Further gene ontology (GO) analysis identified that a large number of defense-related genes, especially in the JA pathway, were specifically down-regulated in *OE-OsNF-YA2* plants infected by RSV (Fig 3F). These data imply that OsNF-YA2 may repress the JA-related pathway in response to RSV infection, powerfully hinting at a role for OsNF-YA2 in the JA-mediated immune response.

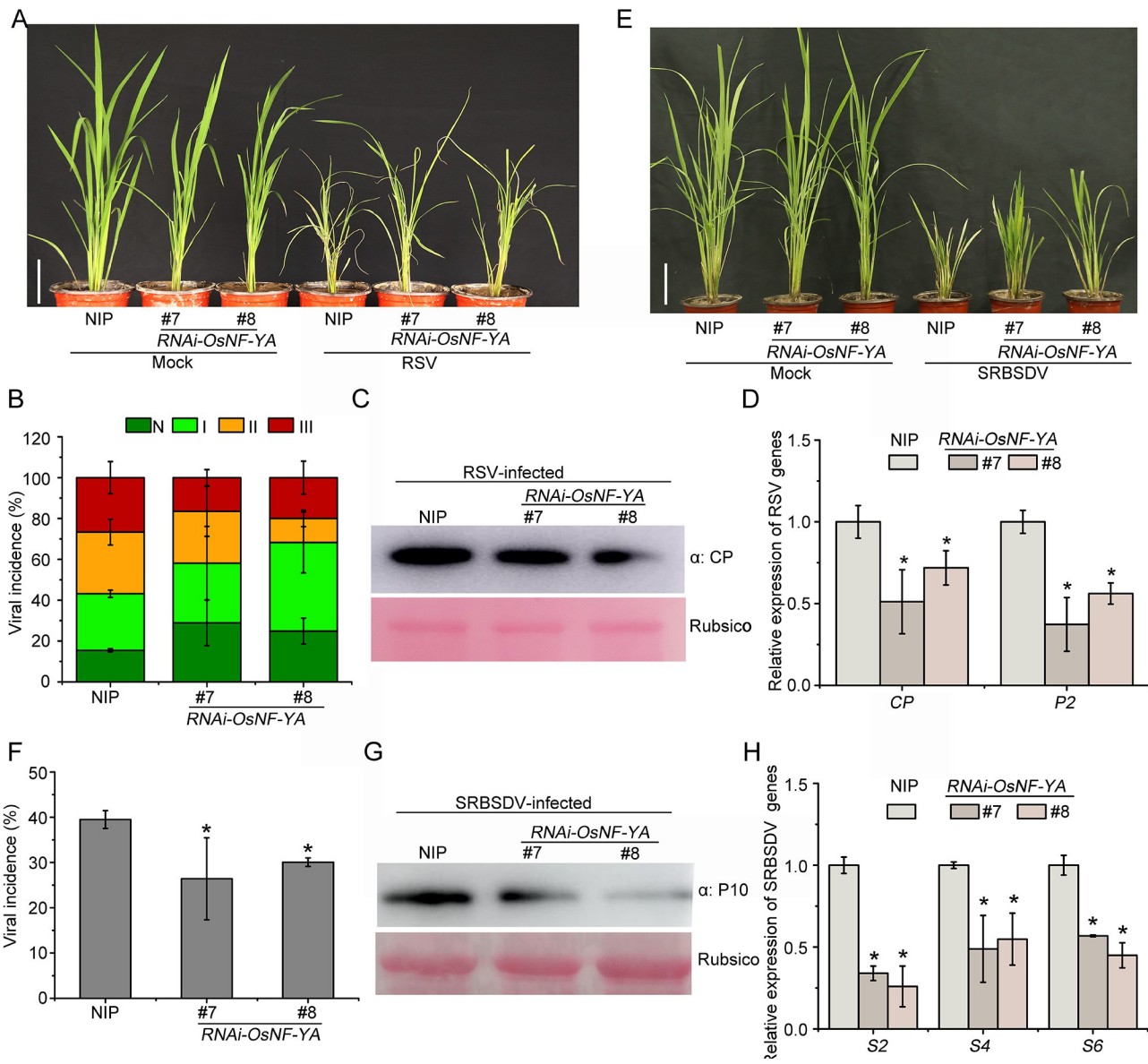

**Fig 2. *OsNF-YAs* RNAi mutants are more resistant to viruses.** A. The symptoms of RSV-infected plants. The phenotypes were observed and photos were taken at 20 dpi. (Scale bars, 10 cm.) B. The percentage of RSV-inoculated plants with different degrees of disease. N, Healthy; I, mild mosaic; II, severe mosaic; III, wilting. C. The accumulation of RSV CP protein in virus-infected plants determined by western blotting. Rubsico was used as an internal reference. D. qRT-PCR results showing the relative expression levels of RSV RNAs (*CP* and *P2*) in virus-infected plants. E. The symptoms of SRBSDV-infected plants. The phenotypes were observed and photos were taken at 30 dpi. (Scale bars, 10 cm.) F. SRBSDV incidence in inoculated *RNAi-OsNF-YA* lines compared with NIP. G. The accumulation of SRBSDV P10 protein in virus-infected plants determined by western blotting, Rubsico was used as an internal reference. H. qRT-PCR results showing the relative expression levels of SRBSDV RNA segments (*S2*, *S4*, *S6*) in virus-infected plants. *OsUBQ5* was used as the internal reference gene to normalize the relative expression. Values shown are the means ± SD of 3 biological replicates. Significant differences were identified using Fisher's least significant difference tests. *At the top of columns indicates significant difference at $p \leq 0.05$.

## OsNF-YAs negatively regulate the JA pathway

To confirm the association of OsNF-YA with the JA pathway, we used qRT-PCR to determine the expression levels of the JA-related genes (*OsJAZ8*, *OsJAZ9*, *OsJAZ11*, *OsJAZ12*, *OsLOX2* and *JiOsPR10*) in *OsNF-YA2* transgenic lines challenged by RSV. In control NIP plants, the

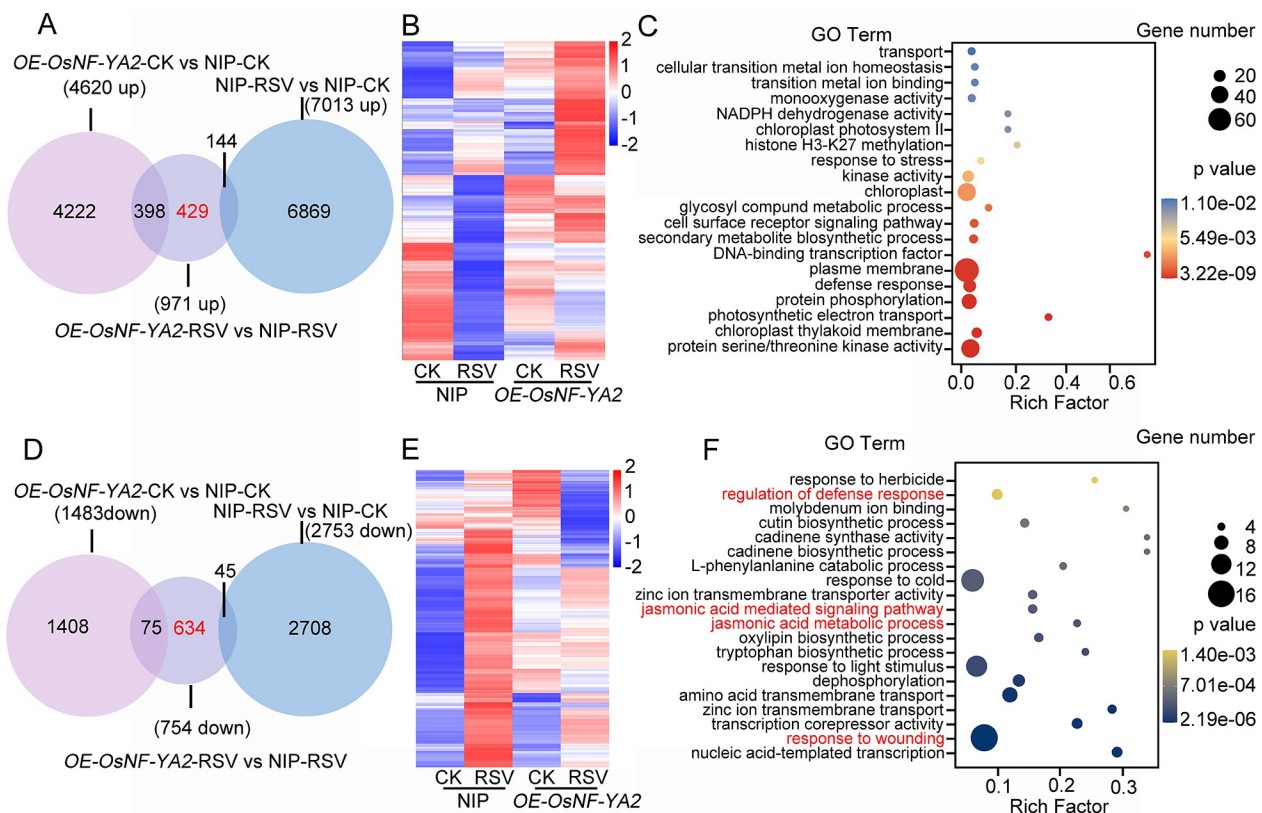

**Fig 3. Transcriptomic analysis of regulatory gene expression profiles in plants over-expressing *OsNF-YA2* and in response to RSV infection.** (**A, D**) Venn diagrams showing the up-regulated (**A**) or down-regulated (**D**) genes resulting from expression of *OsNF-YA2* and infection by RSV. The overlaps indicate the differentially expressed genes in OsNF-YA2-CK versus NIP-CK, OsNF-YA2-RSV versus NIP-RSV, and NIP-RSV versus NIP-CK. (**B, E**) Hierarchical clustering showing the set of 429 genes specifically up-regulated (**B**) or 634 genes specifically down-regulated (**E**) by *OsNF-YA2* under RSV infection. (**C, F**) Gene ontology (GO) showing the up-regulated (**C**) or down-regulated (**F**) genes by *OsNF-YA2* under RSV infection. Numbers indicate the percentages of genes included in each GO category, *p* value display data reliability.

expression of JA-related genes was significantly induced by RSV infection, consistent with previous reports [12,31] (S4A Fig). However, the expression of these JA-related genes was obviously repressed in *OE-OsNF-YA2* plants and activated in *RNAi-OsNF-YA* lines compared with the control NIP plants (Fig 4A). Similar results were obtained when the plants were inoculated with SRBSDV (S4B Fig). These results indicate that OsNF-YA2 negatively regulates the expression of JA-related genes in response to RSV or SRBSDV infection.

We then examined whether OsNF-YA2 affects JA sensitivity by exogenous application of methyl jasmonate (MeJA) to rice in hydroponic culture. The root lengths of wild-type and *OsNF-YA2* transgenic plants were measured 5 days after treatment with 2 or 5 μM MeJA. In the presence of 2 μM MeJA, the root lengths of *OE-OsNF-YA2* lines were less reduced than those of the wild-type plants (40% vs 55%). However, MeJA-treated *RNAi-OsNF-YA* lines had a greater reduction in root length than NIP plants (65% vs 55%) (Fig 4B and 4C). There was a similar effect from 5 μM MeJA treatment. We also performed JA sensitivity assays using stable transgenic rice lines *OE-OsNF-YA1* (#3 and #5) and OE-*OsNF-YA10* (#1 and #2) and found that the root lengths of these *OE-OsNF-YAs* lines were less sensitive to MeJA treatment than those of the wild-type plants (S5 Fig). These results indicate that overexpression of *OsNF-YAs* makes rice plants less sensitive to JA treatment whereas repression of *OsNF-YAs* increases sensitivity and therefore show that OsNF-YAs have negative roles in regulating the JA pathway.

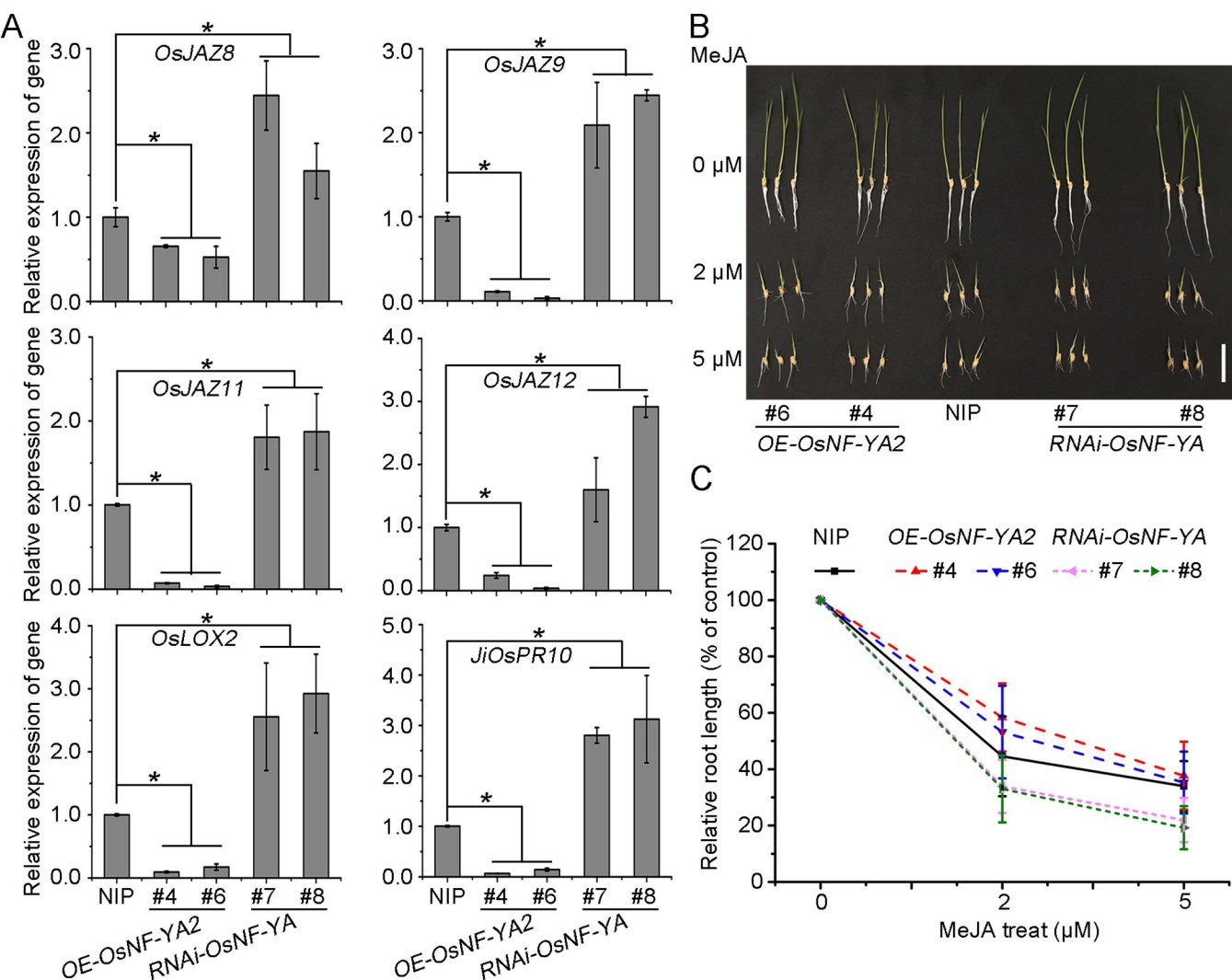

**Fig 4. OsNF-YAs suppress JA signaling.** A. qRT-PCR results showing the relative expression levels of JA pathway genes (*OsJAZ8*; *OsJAZ9*; *OsJAZ12*; *OsJAZ11*; *OsLOX2*; *JiOsPR10*) in *OE-OsNF-YA2* transgenic plants and *RNAi-OsNF-YA* mutant plants compared with NIP background under RSV infection. *OsUBQ5* was used as the internal reference gene to normalize the relative expression. Error bars represent SD of 3 biological replicates. Significant differences were identified using Fisher's least significant difference tests. *At the top of columns indicates significant difference at $p \leq 0.05$. B. Phenotypes of *OE-OsNF-YA2* transgenic plants and *RNAi-OsNF-YA* mutant plants grown on rice nutrient solution containing 2 µM or 5 µM MeJA for 5 days. Scale bar, 5 cm. C. Quantification of root length in indicated plants following MeJA treatment. The root lengths of five-day-old seedlings grown in normal rice culture solutions supplemented with different concentrations of MeJA were measured. Data shown are the means from at least 15 seedlings for each sample. Error bars represent SD. Different letters at the top of columns indicate significant difference a $p \leq 0.05$ by Fisher's LSD tests.

## OsNF-YAs interact with JA key transcription factor OsMYC2/3

Given the negative roles of OsNF-YAs in the JA pathway, we wondered whether OsNF-YAs directly modulate the JA pathway by interacting with the essential components of JA signaling. We have recently shown that JA signaling key transcription factors OsMYC2/3 play vital roles in antiviral immunity against both RSV and SRBSDV [32]. Thus, we hypothesized that OsN-F-YAs might physically associate with OsMYC2/3. To test this hypothesis, we first used a yeast two-hybrid (Y2H) assay which showed that OsMYC2 and OsMYC3, but not OsMYC4, interacted with several OsNF-YA family genes, including OsNF-YA1 and OsNF-YA2 (Fig 5A). Co-immunoprecipitation (Co-IP) assays further confirmed that OsMYC2/3 but not the negative

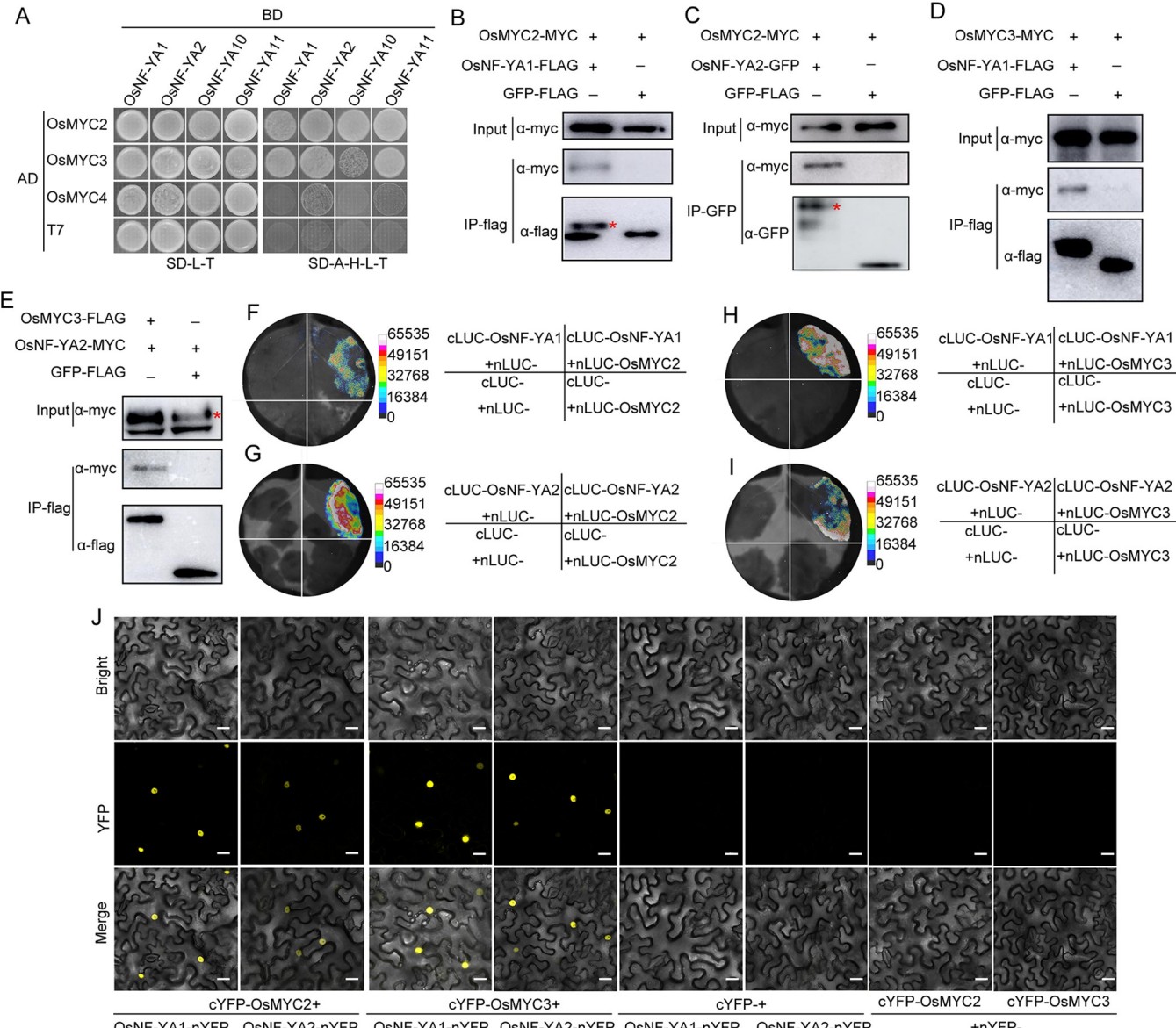

**Fig 5. OsNF-YAs interact with OsMYC2/3.** A. Interaction of OsNF-YAs and OsMYC2/3 proteins in a yeast two-hybrid assay. The CDS for *OsNF-YAs* and *OsMYCs* were introduced into pGBKT7 and pGADT7 vectors, respectively. Yeast cells co-expressing *OsNF-YAs* and *OsMYCs* were grown on selective media SD/-Trp/-Leu (SD-L-T), and interactions were tested with SD/-Trp/-Leu/-His/-Ade (SD-L-T-H-A). Pictures were taken after 3 days incubation at 30˚C. B. Co-IP assays confirming that OsNF-YA1 interacts with OsMYC2. OsMYC2-MYC was co-expressed with OsNF-YA1-FLAG or GFP-FLAG in *N. benthamiana* leaves. Total proteins were extracted and immunoprecipitated by anti-FLAG magnetic beads, and then detected with anti-MYC antibody. C. Co-IP assays confirming that OsNF-YA2 interacts with OsMYC2. OsMYC2-MYC was co-expressed with OsNF-YA2-GFP or GFP-FLAG in *N. benthamiana* leaves. Total proteins were extracted and immunoprecipitated by anti-GFP magnetic beads then detected with anti-MYC antibody. D. Co-IP assays confirming that OsNF-YA1 interacts with OsMYC3. OsMYC3-MYC was co-expressed with OsNF-YA1-FLAG or GFP-FLAG in *N. benthamiana* leaves. Total proteins were extracted and immunoprecipitated by anti-FLAG magnetic beads then detected with anti-MYC antibody. E. Co-IP assays confirming that OsNF-YA2 interacts with OsMYC3. OsMYC3-FLAG was co-expressed with OsNF-YA2-MYC or GFP-FLAG in *N. benthamiana* leaves. Total proteins were extracted and immunoprecipitated by anti-FLAG magnetic beads then detected with anti-MYC antibody. F-I. Results from LCI assays, and the measurements of luciferase activity due to LUC reconstitution for the different combinations, showing that OsNF-YA1 and OsNF-YA2 interact with OsMYC2 and OsMYC3 proteins in *N. benthamiana* leaves. J. BiFC assay showing interactions between OsNF-YA1/OsNF-YA2 and OsMYC2/3 in *N. benthamiana* cells. OsNF-YA1/OsNF-YA2 and OsMYC2/3 were fused with the N-terminal of YFP (nYFP) and the C-terminal of YFP (cYFP), respectively. Scale bar, 50 μm.

control GFP could co-immunoprecipitate with OsNF-YA1 and OsNF-YA2 protein *in vivo* (Fig 5B and 5E). This was further confirmed by a split luciferase complementation imaging (LCI) assay, which showed that co-expression of nLUC-OsMYC2/3 with either cLCU-OsN-F-YA1 or cLUC-OsNF-YA2 resulted in effective luciferase activity in *N. benthamiana* leaves. Neither the combination of nLUC-OsMYC2/3 and cLUC, nor the combination of nLUC- and cLCU-OsNF-YA1/2 activated luciferase activity (Fig 5F–5I). In a bimolecular fluorescence complementation (BiFC) assay, there was a concentrated signal in the plant cell when either OsNF-YA1-nYFP or OsNF-YA2-nYFP was co-expressed with either cYFP-OsMYC2 or cYFP-OsMYC3 in leaves, but there was no YFP signal in the negative control (Fig 5J). Together, these data suggest that OsNF-YAs physically interact with OsMYC2/3.

## OsNF-YAs inhibit the transcriptional activation of OsMYC2/3

We next tested which domain of OsMYC2/3 was responsible for their association with OsN-F-YAs. The transcription factors MYC2/3 usually have a putative transcriptional activation domain (TAD) and a basic-helix-loop-helix domain (bHLH) [33,34]. Based on the conserved sequence characteristics, we constructed several OsMYC3 mutants and found TAD domain was required for the association with OsNF-YA1 and OsNF-YA2 (Fig 6A). Given that the TAD domain is responsible for OsMYC2/3 transcriptional activation, we tested whether OsN-F-YAs affect the transcriptional activation activity of OsMYC2/3 using a dual-LUC reporter system in *N. benthamiana* leaves. The control BD-Ven (BD, which specifically binds the *5×GAL4 cis* element, fused to the fluorescent protein Venus) as the control, BD-OsMYC2, BD-OsMYC3, OsNF-YA1 and OsNF-YA2 act as the effectors (Fig 6B). Consistent with our previous reports [32], BD-OsMYC2 and BD-OsMYC3 have obvious transcriptional activation activity compared with the control BD. However, the transcriptional activation activity of BD-OsMYC2/3 was significantly impaired in the presence of OsNF-YA1 or OsNF-YA2 (Fig 6C–6F). Together, these results suggest that OsNF-YAs inhibit the transcriptional activation of OsMYC2/3.

In the JA signaling pathway, MED25 (Mediator complex subunit 25) is a subunit of the mediating complex, and constitutes the transcription-activating complex with MYC2/3 by binding to the TAD domain [35,36]. Y2H assays showed the TAD motif of OsMYC3 is required for interaction with both OsMED25 and OsNF-YA1/2 (Figs 6A and S6). We then hypothesized that OsNF-YAs might compete with OsMED25 to bind OsMYC2/3. We first designed protein competition Co-IP assays in *N. benthamiana*. Immunoblot analysis indicated that OsMED25 can bind to OsMYC2/3 when co-infiltrated with the negative control HA-GFP, but the association between OsMED25 and OsMYC2/3 was significantly reduced in the presence of OsNF-YAs (Fig 6G–6J). A BiFC experiment was also used as a protein competition assay in *N. benthamiana* leaves. The fluorescence in the combinations of OsMED25-nYFP and cYFP-OsMYC2/3 was observed in presence of the control Gus-myc, but was obviously weakened in the presence of OsNF-YAs (S7A and S7B Fig). In addition, we then conducted a split luciferase complementation imaging assay which showed that the co-expression of either nLUC-OsMYC2 or nLUC-OsMYC3 with cLUC-OsMED25 resulted in effective luciferase activity in *N. benthamiana* leaves when co-infiltrated with the negative control HA-GFP, but that this activity was reduced in the presence of OsNF-YAs (S7C and S7D Fig). Results of qRT-PCR assays showed that there were no significant differences in transcript expression of MYC2/3 and MED25 in these BiFC and LCI assays (S8 Fig). Taken together, these results show that OsNF-YAs repress JA signaling by directly disturbing the OsMYC2/3-OsMED25 complex and impeding the transcriptional activation activity of OsMYC2/3.

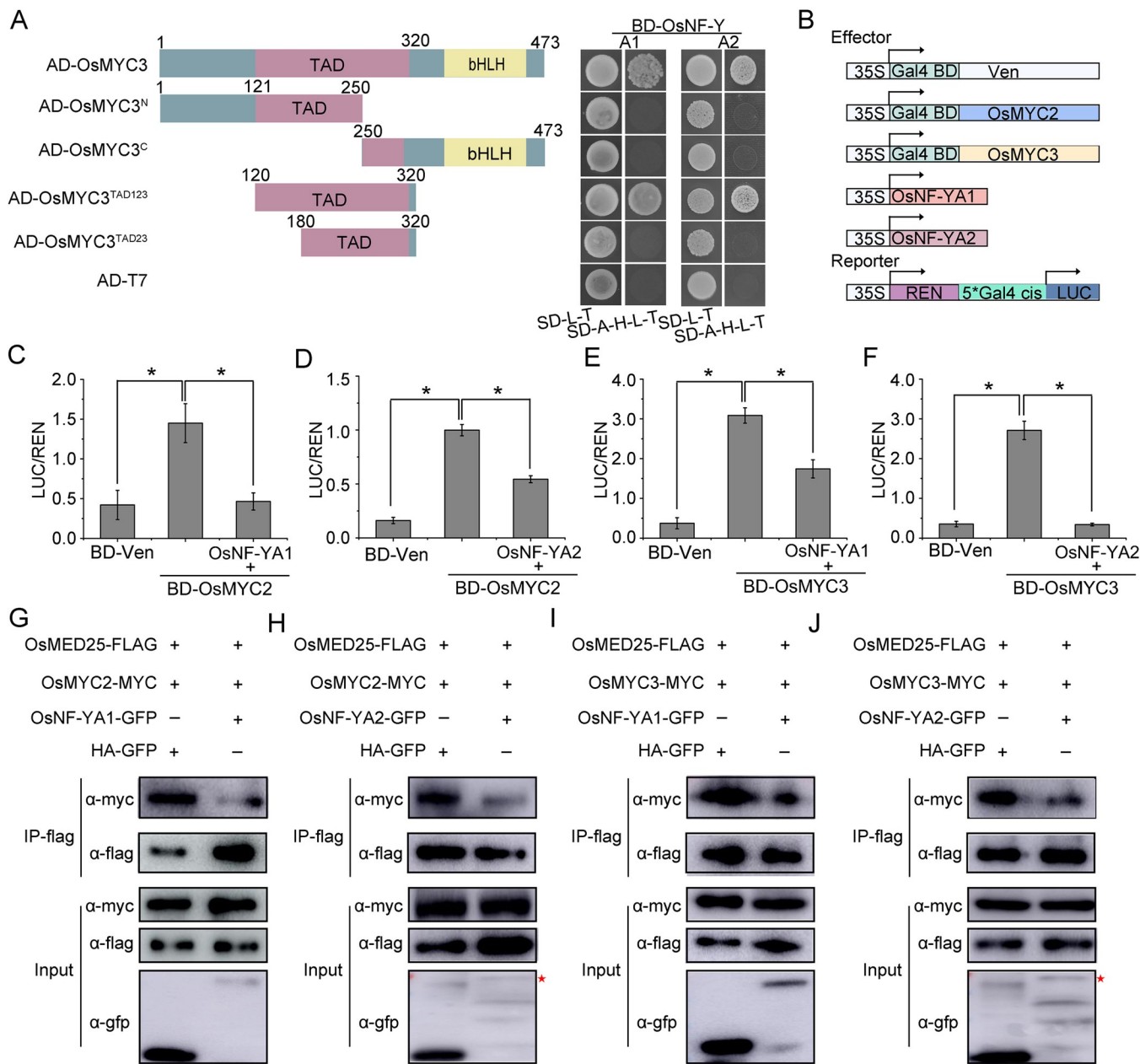

**Fig 6. OsNF-YAs disturb the transcriptional activation of OsMYC2/3.** A. Interaction domain of OsMYC3 mutants and OsNF-YA1 proteins in a yeast two-hybrid assay. The CDS for OsNF-YA1 and OsMYC3 mutants were introduced into pGBKT7 and pGADT7 vectors, respectively. Yeast cells co-expressing plasmid were grown on selective media SD/-Trp/-Leu (SD-L-T), and interactions were tested with SD/-Trp/-Leu/-His/-Ade (SD-L-T-H-A). Pictures were taken after 3 d of incubation at 30°C. B. Schematic diagrams of the effectors and reporters used in the dual-LUC experiments. The effectors were BD, OsMYC3-BD, and OsNF-YAs fused MYC-tag. The reporters were 35S: REN-*5*gal* pro:LUC plasmids. The BD domain can bind the *5*gal* promoter. C-F. The relative LUC activities measured in *N.benthamiana* leaves, using the combinations shown in B. The empty BD effector was used as a control. The LUC/REN ratio represents the relative LUC activity. Different letters at the top of columns indicate significant difference at $p \leq 0.05$ by Fisher's LSD tests. G-J. The effect of OsNF-YAs on the interaction between OsMYC2/3 and OsMED25 by Co-IP assays. The combinations of OsMED25-flag and OsMYC2/3-myc with or without OsNF-YA1-GFP/OsNF-YA2-GFP were co-infiltrated into leaves of *N. benthamiana*. Total proteins were extracted and immunoprecipitated by anti-FLAG magnetic beads. The immunoprecipitated fractions were probed with anti-flag, anti-myc, and anti-GFP antibodies.

## Discussion

The NF-Y transcription factors complex is composed of three subunits, NF-YA, -YB and -YC, and NF-YA plays diverse roles in a variety of developmental processes including plant growth, development and stress responses [37]. For instance, lines overexpressing *AtNF-YA5* were more resistant to drought, while nf-ya5 lines were hypersensitive to drought in comparison with wild-type plants [38]. In rice, overexpression of *OsNF-YA7*, but not *OsNF-YA4*, mediates drought tolerance in an ABA-independent manner, suggesting an important role for OsNF-YA7 in the water acquisition signaling pathway [28]. In *Arabidopsis*, lines overexpressing *AtNF-YA3*, *AtNF-YA7* and *AtNF-YA10* were more sensitive to ABA [39]. In wheat (*Triticum aestivum*), overexpression of *TaNF-YA10* increased sensitivity to salt stress [40]. While numerous studies have reported that OsNF-YAs participate in plant growth and developmental processes, their role(s) in plant resistance to pathogens have rarely been reported. Recently, overexpression of *miR169a* and suppression of its target *OsNF-YAs* was shown to enhance rice susceptibility to *M. oryzae* by reducing hydrogen peroxide accumulation, indicating that OsNF-YAs positively modulate rice defense against fungal pathogens [30]. However, it is unclear whether *OsNF-YAs* have similar roles in the interactions between rice and viruses. Here, we showed that the expression of most *OsNF-YAs* was induced when rice was infected by either of two distinct viruses, RSV and SRBSDV (Fig 1A and 1B). Overexpression of three representative *OsNF-YAs* increased the susceptibility of rice to both viruses while *OsNF-YAs* RNAi mutant plants were more resistant than the controls (Figs 1 and 2 and S3). These results suggest that, in contrast to their positive roles in plant defense against fungal pathogens, *OsNF-YAs* negatively regulate rice antiviral immunity.

Recent evidence has shown that NF-Y family members associate with plant hormone pathways to regulate plant growth development and stress responses. *AtNF-YCs* negatively regulate BR biosynthesis genes by stabilizing BIN2 to inhibit light-induced hypocotyl elongation [41]. Simultaneously, *AtNF-YCs* mediate seed germination by interacting with the ABA and GA pathway [42]. AtNF-YCs interact with the master regulators of ABA signaling ABF3 and ABF4 to regulate the expression of the *SOC1* gene and accelerate flowering under drought stress in *Arabidopsis thaliana* [43]. *AtNF-YA2* and *AtNF-YA10* participate in the regulation of *Arabidopsis* leaf growth and development by directly binding to *cis*-CCAAT in the *YUC2* promoter to inhibit auxin biosynthesis [44]. However, little has been reported about the direct interplay between NF-Y family proteins and the JA pathway. Here, transcriptome analysis and qRT-PCR showed that, in response to infection by rice viruses, the JA pathway was down-regulated when *OsNF-YA2* was overexpressed but up-regulated in *OsNF-YA* RNAi mutants (Figs 3D–3F, 4 and S4B). Overexpression of *OsNF-YAs* made plants less sensitive to JA treatment (Figs 4A and S5). JA treatment significantly enhanced the resistance of wild-type rice plants to RSV infection compared to *OE-OsNF-YA2* transgenic plants (43% VS 12%) (S9A Fig). qRT-PCR and Western blotting experiments confirmed that JA treatment significantly reduced the accumulation of RSV CP level in wild-type rice plants (51%), while the extent of JA treatment was much alleviated in *OE-OsNF-YA2* transgenic plants (28% or 25%) (S9B and S9C Fig). These results indicated that OsNF-YAs make rice insensitive to JA treatment and inhibit JA-mediated antiviral defense. OsNF-YAs were shown to physically interact with OsMYC2/3 (Fig 5), directly disturbing the OsMYC2/3-OsMED25 complex by competing for the TAD domain (Figs S6 and 6G–6J). Together, our results showed that OsNF-YAs participate in and repress JA signaling by directly interacting with and impeding the transcriptional activity of OsMYC2/3.

Phytohormones play critical roles in various aspects of plant growth regulation and defense response. In rice, auxin and the JA pathway participate positively in defense against viruses,

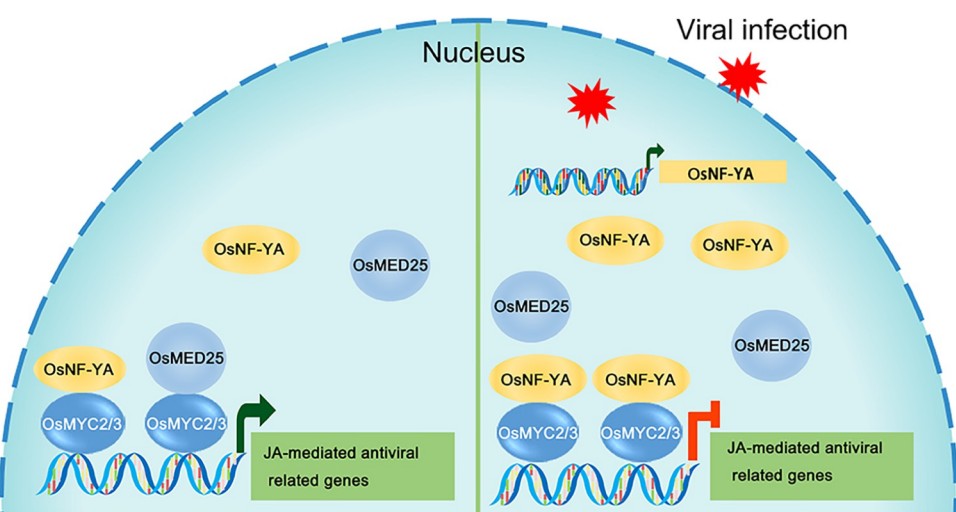

**Fig 7. Model showing how OsNF-YAs manipulate the JA-mediated antiviral defense.** In healthy plants, some OsNF-YAs combine with OsMYC2/3 transcription factors and slightly affect the JA-mediated antiviral defense. In virus-infected plants, OsNF-YAs are markedly induced by RSV or SRBSDV and dissociate the OsMYC2/3-OsMED25 complex, inhibiting the JA-mediated antiviral defense response, favoring virus infection.

the BR pathway negatively regulates resistance to RBSDV by interfering with the JA pathway and ABA inhibits JA-mediated antiviral defense [8–11,32,45,46]. Thus, the various synergistic or antagonistic interactions between phytohormones play important roles in the outcome of antiviral responses. To successfully infect plants, rice viruses encode pathogenicity factors that target the essential components of antiviral phytohormones to achieve infection. For example, we recently showed that different proteins encoded by viruses interact with auxin response factor OsARF17 and weaken the auxin-mediated antiviral defense response [46]. These different viral proteins can also hijack and repress the key components of the JA pathway (OsJAZs and OsMYC2/3), benefiting virus pathogenicity and vector transmission [32]. Recently, RSV coat protein (CP) was shown to act as an elicitor to trigger JA accumulation, and JA signaling synergized with RNA silencing to promote antiviral defense. These studies suggest that the JA pathway plays essential roles in the battlefield between rice and viruses. Our work here has identified some new regulators of the JA pathway, OsNF-YAs, which directly associate with the TAD domain of OsMYC2/3 disturb the OsMYC2/3-OsMED25 complex and repress their transcriptional transactivation.

Whereas OsNF-YAs have been shown to positively regulate plant defense against fungal and bacterial pathogens, our results show that they negatively impact defense against viruses by interfering with the JA-mediated antiviral defense. We propose a model in which the enhanced expression of OsNF-YAs following virus infection leads OsNF-YAs to interact with OsMYC2/3 transcription factors. This impedes their transcriptional activation activity by interfering with the transcriptional activation complex OsMYC2/3-OsMED25, finally resulting in repression of JA-mediated antiviral immunity (Fig 7).

## Materials and methods

### Plant materials and growth conditions

All rice plants were grown in the greenhouse at 28–30˚C under 10 h light/14 h dark. To generate transgenic plants over-expressing *OsNF-YAs*, the full length of these genes was cloned and the expression vector was constructed by homologous arm recombination. The recombinant

plasmid was electroporated into *Agrobacterium tumefaciens* strain *GV3101* and transformed into *O. sativa* L. *japonica* Nipponbare (NIP) background. *RNAi-OsNF-YA* lines with silenced *OsNF-YAs* expression were also constructed in the NIP background. *Nicotiana benthamiana* plants were grown in a growth chamber at 25˚C and with 16 h light/8 h darkness. The primers used are listed in S1 Table.

### Insect vectors and virus inoculation assay

The rice virus inoculation methods have been detailed in our previous reports [11,32,46]. For RSV inoculation, 2–3 instar larvae of SBPH were allowed to feed on rice seedlings at the 4-leaf stage for 3 days, and then removed. The inoculated plants were then grown in the greenhouse for observation and were tested at 30 dpi by qRT-PCR and western blotting. Similar methods were used for SRBSDV inoculation. The specific primers used to detect SRBSDV/RSV are listed in S1 Table. The number of infected plants was determined following RT-PCR to calculate the viral incidence. All experiments were repeated at least three times with similar results.

### Hormone treatments

Stock solutions of MeJA (Sigma) in 100% ethanol were diluted with sterile distilled water containing 0.1% Triton X-100. RSV-inoculated rice seedlings were then sprayed with 50 μM MeJA or 0.1% Triton X-100 as the mock control. Each treatment used at least 30 seedlings. Disease symptoms were observed 20 days post inoculation (dpi). Leaf samples were frozen in liquid nitrogen and stored at −80˚C for protein and RNA extraction.

### Total RNA extraction and quantitative real-time PCR (qRT-PCR) analysis

Total RNA was extracted from rice leaves using the TRIzol reagent (Invitrogen, Carlsbad, CA, USA, Cat. no. 15596–026) according to the manufacturer's instructions. 1 μg of total RNA was mixed with 4*gDNA wiper mix to eliminate genomic DNA at 42˚C for 2 minutes and then reverse transcribed to cDNA using 5* HiScript III qRT Super Mix (Vazyme). qRT-PCR analysis was performed using the Hieff qPCR SYBR Green Master Mix (Low ROX Premixed) by the ABI7900HT Sequence Detection System (Applied Biosystems, Carlsbad, CA, USA). The reference gene *OsUBQ5* (AK061988) was used to normalize the relative expression levels [47], and the method of $2^{-\Delta\Delta Ct}$ was used for analysis [48]. The experiments were repeated at least three times. Each biological sample consisted of more than 10 plants. The qRT-qPCR primers used in this study are listed in S1 Table.

### Western blot analysis

Total proteins were extracted from infected rice leaves with 10% SDS lysis buffer (0.1M Tris-HCl (pH = 6.8), 10% SDS, 2% β-mercaptoethanol) and separated in 12% SDS-PAGE gels before transfer to a methanol pre-activated PVDF membrane. This was then incubated for 2h at room temperature with anti-CP or anti-P10 (provided by Prof. Jianxiang Wu) monoclonal antibodies to specifically recognize RSV and SRBSDV, respectively. After incubating with IgG-HRP antibody, the protein membranes were imaged using ECL substrate by the BIO-RAD ChemiDoc MP Imaging System.

### RNA-seq analysis

The methods of RNA library construction were described in our previous reports [8,49]. Briefly, RNA sequencing used three samples each of mock and RSV-infected rice plants at 20 dpi. Total RNA was extracted using TRIzol reagent (Invitrogen, Carlsbad, CA, USA, Cat. no.

15596–026), the RNA quantity and purity were determined by Hangzhou Lianchuan (Hangzhou, China) and the Illumina HiSeq 2000 platform was used for RNA sequencing. The rice genome (MSU Rice Genome Annotation Project database version 7.0) was used for mapping of sequencing reads by Bowtie software. Blast2go software was used for the GO functional classes and KEGG pathways. Significant differential expression of genes was assessed using the absolute value of $\log_2$ (fold change) ratio $\geq 1$ and $p \leq 0.05$ in Tukey's multiple comparison test.

### Root growth inhibition assay

Seeds ($\geq 20$ seeds per line) were germinated at 37˚C and then grown in rice nutrient solution plant hydroponic boxes containing different concentrations of MeJA (2 and 5 μM) with a regime of 8 h light at 25˚C/16 h dark at 30˚C. Five days later, the root lengths were measured and recorded, and the relative root length was used to evaluate the sensitivity to JA. Each measurement was repeated three times.

### Yeast Two-Hybrid Assay

Full-length coding sequences of the *OsNF-YAs* (*OsNF-YA1*, *OsNF-YA2*, *OsNF-YA10*, *OsNF-YA11*) genes were amplified by PCR using the primers listed in S1 Table and inserted into the bait vector pGBKT7. The full-length coding sequences of the *OsMYCs* (*OsMYC2/3/4*) and their different truncated variants were described in our previous reports [32]. Different combinations of plasmids were co-transformed into the yeast strain AH109 according to the product specification (Takara, Japan). The transformants were cultivated on SD/-Leu/-Trp plates for about 3 days, and then positive clones were transferred to SD/-Leu/-Trp/-His/-Ade plates for interaction tests. Yeast cells were photographed after 3 days at 30˚C to record growth. All experiments were repeated three times with similar results.

### Co-immunoprecipitation (Co-IP) assay

*N. benthamiana* leaves (four-weeks-old) were co-infiltrated by agro-infiltration. After 48 hours, the leaf tissue was collected, quick-frozen with liquid nitrogen and ground into powder. Total proteins were extracted using cold IP buffer (40 mM Tris-HCl (pH = 7.5), 100 mM NaCl, 0.5% Triton X-100, 1 mM EDTA, 1% glycerol with 1 mM DTT, 1 mM PMSF) for 20 min on ice and centrifuged twice at highest speed for 10 min at 4˚C. Then, 60μl was used as input control for western blot and the remaining supernatant was transferred to a new microcentrifuge tube, where it was incubated with 5μl FLAG-trap beads or GFP-trap beads (Sigma-Aldrich, USA) at 4˚C for approximately 2 h. The supernatant was then removed and the beads were washed at least three times in ice-cold IP buffer to remove the non-specifically adsorbed proteins. Subsequently, all protein samples were boiled for 10 minutes, briefly centrifuged and then separated on a 12% SDS-PAGE gel followed by western blotting analysis.

### BiFC analysis

The coding sequences of *OsNF-YA1/2*, *OsMYC2/3* were amplified and cloned into the nYFP and cYFP vectors, respectively. The primers used are listed in S1 Table. Equal amounts of the plasmids were co-transformed into *Agrobacterium* strain GV3101 and then a combination of *Agrobacterium* samples, each with $OD_{600} = 0.4$, was co-injected into *N. benthamiana*. The YFP fluorescence signals were detected under a confocal laser scanning microscope (Nikon). Three biological repeats were done with similar results.

### Luciferase complementation imaging (LCI) assay

For the LUC complementation assays, the coding sequences of *OsNF-YA1/2* and *OsMYC2/3* were combined into *pCAMBIA1300-cLUC* and *pCAMBIA1300-nLUC* vectors, respectively. Equal amounts of the plasmids were co-transformed into *Agrobacterium* strain GV3101 and then a combination of *Agrobacterium* samples, each with $OD_{600}$ = 0.4, was co-infiltrated into different areas of the same *N. benthamiana* leaf. The plants were cultivated in the greenhouse for 48h, and then 0.1 mM luciferase substrate was infiltrated into the same positions for 10 minutes. Imaging was performed in a low-light CDD imaging apparatus (LUMAZONE SOPHIA2048B, USA) with pre-cooling. All the experiments were independently repeated at least 3 times and produced similar results.

### Dual luciferase transient transcriptional activity assay

For dual-LUC determination, empty pCV-BD was used as a vector to link with the coding sequences of *OsMYC2* and *OsMYC3*. OsNF-YA1-MYC and OsNF-YA2-MYC were used as effectors. The $5^*gal$ promoter was used to drive the firefly luciferase gene (LUC) as a reporter and the Renillia luciferase (REN) reporter gene controlled by Cauliflower mosaic virus promoter (35S) was the reference. The BD domain can bind to the $5^*gal$ promoter. All plasmids were transformed into *Agrobacterium* strain GV3101, and then a combination of *Agrobacterium* samples, each at the concentration of $OD_{600}$ = 1.0, was co-infiltrated into different areas of the same *N. benthamiana* leaf. The detection methods were described in our previous reports [32]. The relative luciferase activity was analyzed used LUC/REN ratios. All the experiments were repeated more than 3 times to calculate the Mean ± s.d. of relative ratio of LUC/REN. The primers used for these constructs are listed in S1 Table.

### Statistical analysis

Differences were analyzed using one-way or two-way ANOVA with Fisher's least significant difference tests. A $p$-value $\leq$ 0.05 was considered statistically significant. All analyses were performed using ORIGIN 9.1 software.

## Supporting information

**S1 Fig. The expression levels of *OsNF-YAs* genes in *OsNF-YAs* transgenic plants and the survival rates of the insect vectors on them.** The relative expression levels of *OsNF-YAs* genes determined in RNA extracted from *OE-OsNF-YAs* (A) or *RNAi-OsNF-YAs* (B) plants at 15 dpi. Data are shown as relative expression levels of transgenic plants in comparison with mock plants. C and D. The survival rates of small brown planthoppers (SBPH) and white-backed planthoppers (WBPH) on NIP and *OsNF-YAs* transgenic plants. Ten-day-old seedlings of *OsNF-YAs* transgenic lines were infested with virus-free SBPH (C) and WBPH (D). 3 insects were placed on each seedling for 5 days and the numbers surviving were counted 3 and 5 days later.
(TIF)

**S2 Fig. The symptoms of RSV-infected plants with different disease symptom grades.** According to the severity of the symptoms, we classified RSV infected plants into four grades: healthy (N), mild mosaic (I), severe mosaic (II), wilting (III).
(TIF)

**S3 Fig. OsNF-YA10 negatively modulates rice resistance to RSV and SRBSDV.** A. The symptoms of RSV infection in *OE-NF-YA10* transgenic lines and NIP controls. The

phenotypes were observed and photos were taken at 20 dpi. (Scale bars, 5 cm.) B. The percentage of RSV-infected plants with different degrees of disease. N, Healthy; I, mild mosaic; II, severe mosaic; III, wilting. C. The accumulation of RSV CP protein in RSV-infected plants determined by western blotting. Rubsico was used as the internal reference. D. qRT-PCR results showing the relative expression levels of RSV RNAs (*CP* and *P2*) in virus-infected plants. E. The symptoms of SRBSDV infection at 30 dpi. (Scale bars, 5 cm.) F. Viral incidence in *OE-NF-YAs* transgenic lines compared with NIP controls. G. The accumulation of SRBSDV P10 protein in virus-infected plants determined by western blotting. Rubsico was used as the internal reference. H. qRT-PCR results showing the relative expression levels of SRBSDV RNA segments (*S2, S4, S6*) in virus-infected plants. *OsUBQ5* was used as the internal reference gene to normalize the relative expression. Values shown are the means ± SD of 3 biological replicates. Significant differences were identified using Fisher's least significant difference tests. *At the top of columns indicates significant difference at $p \leq 0.05$.
(TIF)

**S4 Fig. OsNF-YAs suppress JA signaling.** A. Hierarchical clustering showing the changed JA pathway genes in NIP-CK and NIP-RSV. B. qRT-PCR results showing the relative expression levels of JA pathway genes (*OsJAZ8; OsJAZ9; OsJAZ11; OsJAZ12; OsLOX2; JiOsPR10*) in SRBSDV-infected plants compared with NIP background from three biological replicates in a one-way ANOVA and evaluated at $p \leq 0.05$ by Fisher's least significant difference tests. *OsUBQ5* was used as the internal reference gene to normalize the relative expression. Error bars represent SD of 3 biological replicates. Significant differences were identified using Fisher's least significant difference tests. *At the top of columns indicates significant difference at $p \leq 0.05$.
(TIF)

**S5 Fig. Root length sensitivity of *OsNF-YAs* transgenic plants following MeJA treatment.** A and C. Phenotypes of *OE-OsNF-YA1* (A) and *OE-OsNF-YA10* (C) transgenic plants grown on rice nutrient solution containing 2 µM or 5 µM MeJA for 5 days. Scale bar, 2 cm. B and D. Root lengths of *OE-OsNF-YA1* (B) and *OE-OsNF-YA10* (D) transgenic plants following MeJA treatment. The root lengths of five-day-old seedlings grown in normal rice culture solutions supplemented with different concentrations of MeJA were measured. Data shown are the means from at least 15 seedlings for each sample. Error bars represent SD. Different letters at the top of columns indicate significant difference a $p \leq 0.05$ by Fisher's LSD tests.
(TIF)

**S6 Fig. TAD motif of OsMYC3 is required for interaction with OsMED25 or OsNF-YAs.** Interaction of OsMED25 and OsMYC3 mutant proteins in a yeast two-hybrid assay. The CDS for *OsNF-YAs* and *OsMED25* were introduced into pGBKT7 and OsMYC3 mutants were introduced into pGADT7 vectors, respectively. Yeast cells co-expressing OsNF-YAs or OsMED25 and OsMYC3 mutants were grown on selective media SD/-Trp/-Leu (SD-L-T), and interactions were tested with SD/-Trp/-Leu/-His/-Ade (SD-L-T-H-A). Pictures were taken after 3 days incubation at 30°C.
(TIF)

**S7 Fig. OsNF-YAs interfere with the association between OsMYC2/3 and OsMED25.** A-B. BiFC assays showing that OsNF-YA1/2 interfere with the interaction between OsMED25 and OsMYC2/OsMYC3. Fusion proteins were transiently expressed in leaves of *N. benthamiana* and observed by confocal microscopy. The YFP signals were reduced in the presence of OsNF-YA1/2. (Scale bar, 100 µm). Quantification of the fluorescence intensity from leaves co-expressing cYFP-OsMYC2 or cYFP-OsMYC3 and OsMED25-nYFP with Gus-myc or

OsNF-YA1/2. Data are means from 40 to 60 transfected cells. Error bars represent SD. Significant differences were identified using Fisher's least significant difference tests. *At the top of columns indicates significant difference at p ≤ 0.05. C, D. LCI assays showing that OsNF-YA1/2 interferes with the interaction between OsMED25 and OsMYC2/OsMYC3. Fusion proteins were transiently expressed in leaves of *N. benthamiana* and the measurements of luciferase activity due to LUC reconstitution for the different combinations. Quantification of the fluorescence intensity from leaves co-expressing cLUC-OsMED25 and OsMYC2-nLUC or OsMYC3-nLUC with HA-GFP or OsNF-YA1/2. Error bars represent SD of 3 biological replicates. Significant differences were identified using Fisher's least significant difference tests. *At the top of columns indicates significant difference at p ≤ 0.05.
(TIF)

**S8 Fig. The effect of OsNF-YAs on the transcriptional expression of *OsMYC2/3* or *OsMED25* in *N. benthamiana*.** A-F. qRT-PCR results showing the transcript expression of *OsMYC2/3* and *OsMED25* in these BiFC and LCI assays. The relative expression levels of genes (*OsMED25; OsMYC2; OsMYC3*) in these BiFC (A-C) and LCI (D-F) assays using three biological replicates in a one-way ANOVA and evaluated at p ≤ 0.05 by Fisher's least significant difference tests. NbUBC was used as the internal reference gene to normalize the relative expression. Error bars represent SD of 3 biological replicates.
(TIF)

**S9 Fig. Effects of exogenous JA on RSV-infected *OE-OsNF-YA2* transgenic lines.** A. Viral incidence in *OE-OsNF-YA2* transgenic lines and NIP controls under JA treatment. B. qRT-PCR results showing the relative expression levels of RSV *CP* gene in virus-infected plants. *OsUBQ5* was used as the internal reference gene to normalize the relative expression. Values shown are the means ± SD of 3 biological replicates. Significant differences were identified using Fisher's least significant difference tests. *At the top of columns indicates significant difference at p ≤ 0.05. C. The accumulation of RSV CP protein in virus-infected plants determined by western blotting. Rubsico was used as the internal reference.
(TIF)

**S10 Fig. Original images for immunoblots.**
(TIF)

**S1 Table. The primers used in the article.**
(XLSX)

**S2 Table. The differentially up-regulated genes in OE-OsNF-YA2-CK versus NIP-CK, OE-OsNF-YA2-RSV versus NIP-RSV, and NIP-RSV versus NIP-CK subsets.**
(XLS)

**S3 Table. Expression (FPKM) of 429 candidate genes among 971 genes specifically up-regulated in *OE-OsNF-YA2* transgenic plants after RSV infection.**
(XLS)

**S4 Table. The differentially down-regulated genes in OE-OsNF-YA2-CK versus NIP-CK, OE-OsNF-YA2-RSV versus NIP-RSV, and NIP-RSV versus NIP-CK subsets.**
(XLS)

**S5 Table. Expression (FPKM) of 634 candidate genes among 754 genes specifically down-regulated in *OE-OsNF-YA2* transgenic plants after RSV infection.**
(XLS)

## Acknowledgments

We thank Prof. Jianxiang Wu (Zhejiang University) for providing viral proteins antibody. We thank Prof. Guohui Zhou and Dr. Tong Zhang for providing SRBSDV-infected plants. We thank Prof. Qingyun Bu (Chinese Academy of Sciences) for providing OsMED25 plasimid. We thank Mike Adams for critically reading and improving the manuscript.

## Author Contributions

**Conceptualization:** Zongtao Sun.

**Formal analysis:** Zongtao Sun.

**Funding acquisition:** Hehong Zhang, Jianping Chen, Zongtao Sun.

**Investigation:** Xiaoxiang Tan, Hehong Zhang, Zihang Yang, Zhongyan Wei, Yanjun Li, Zongtao Sun.

**Methodology:** Xiaoxiang Tan, Hehong Zhang, Zihang Yang, Zhongyan Wei, Yanjun Li.

**Project administration:** Jianping Chen.

**Supervision:** Jianping Chen, Zongtao Sun.

**Writing – original draft:** Xiaoxiang Tan, Zongtao Sun.

**Writing – review & editing:** Jianping Chen, Zongtao Sun.

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
