## [Decision Letter · Decision Letter 0]

14 Feb 2022

Dear Dr. Sun,

Thank you very much for submitting your manuscript "NF-YA transcription factors suppress jasmonic acid-mediated antiviral defense for facilitating viral infection in rice" for consideration at PLOS Pathogens. As with all papers reviewed by the journal, your manuscript was reviewed by members of the editorial board and by several independent reviewers. In light of the reviews (below this email), we would like to invite the resubmission of a significantly-revised version that takes into account the reviewers' comments.

Both reviewer #1 and #3 have raised issues with the conclusion that OsNF-YAs inhibit the transcriptional activation of OsMYC2/3 by direct association with the TAD domain. Additional experiments suggested by the reviewers should be conducted to resolve these issues. Furthermore, quantitative immunoprecipitation experiments should be included to demonstrate that OsNF-YA disrupts the interaction between  OsMYCs and OsMED25 through the TAD domain. The influence of JA on the RSV and SRBSDV resistance using OsNF-YA2 RNAi and overexpression lines as suggested by reviewer #2 is important to show that NF-YA2 suppresses JA signal to promote viral pathogenesis.

We cannot make any decision about publication until we have seen the revised manuscript and your response to the reviewers' comments. Your revised manuscript is also likely to be sent to reviewers for further evaluation.

Sincerely,

Savithramma P. Dinesh-Kumar

Associate Editor

PLOS Pathogens

Peter Nagy

Section Editor

PLOS Pathogens

Kasturi Haldar

Editor-in-Chief

PLOS Pathogens

orcid.org/0000-0001-5065-158X

Michael Malim

Editor-in-Chief

PLOS Pathogens

orcid.org/0000-0002-7699-2064

Reviewer's Responses to Questions

**Part I - Summary**

Reviewer #1: Rice viral diseases are the major threats to rice production and food security worldwide. Among them, RSV and SRBSDV are two important viruses in Asia. In this manuscript, Tan et al showed evidence that RSV and SRBSDV infection both induced the expression of OsNF-YA family genes, which inhibit JA mediated antiviral defense by dissociating the OsMYC2/3 - OsMED25 complex to facilitate virus infection. Overall, the story is interesting and novel. However, there are weakness and mistakes that need to be addressed before acceptance for publication.

Major:

1) The authors claim that OsNF-YAs inhibit the transcriptional activation of OsMYC2/3 by direct association with the TAD domain, but the evidence is not convincing. Firstly, Fig6G - Fig6J can`t illustrate the OsNF-YA disrupt the interaction of OsMYCs and OsMED25 through TAD domain, the MYCTAD should be added to the experiment system; Secondly, BiFC and LCI assays are both imaging assays, quantitative analysis should be carried out to directly prove their interaction is disrupted. Besides, the authors should further confirm this conclusion in corresponding transgenic materials.

2) Both RSV and SRBSDV infection induce the expression of the 11 OsNF-YAs, and the authors constructed the overexpression lines of OsNF-YA1, OsNF-YA2 and OsNF-YA10, what`s the reason for choosing OE-OsNF-YA2 lines to perform RNA-seq and JA sensitivity assays?

Minor:

3) The authors concentrate solely on the expression level of OsNF-YAs after virus infection and the subsequent experiments, since OsNF-Y is a heterotrimeric complex consisting of OsNF-YA, OsNF-YB and OsNF-YC subunit, did OsNF-YB and OsNF-YC also play roles in antiviral defense? How do they respond to virus infection? Do they interact with OsMYC2/3?

4) The authors find OsNF-YAs negatively regulate JA pathway by qRT-PCR and JA sensitivity assays indirectly, it would be better to measure the JA contents in OE-OsNF-YA lines or Os-NF-YA RNAi mutant lines.

5) In the yeast two hybrid assay in Fig6A, MYC3N also obtain a TAD domain, it is strange that MYC3N can`t interact with OsNF-YA1 since MYC3TAD123 can interact with OsNF-YA1. Also, it would be better to label the schematic diagram in the left.

6) I don`t think the photograph of Grade II in FigS2 shows the typical symptoms of RSV infected plants.

7) In the dual-LUC reporter system in Fig6B, the authors use BD empty vector as control, they should choose a better control which can express a certain protein instead of an empty vector.

8) The authors use BiFC and LCI assays to show OsNF-YAs can imped the interaction between MYC2/3 and MED25 in Fig6.G-J. Nevertheless, they should carry out western blotting assay to exclude the effect of protein expression level.

9) Ponceau S staining in Fig1E shows the loading is not equal, which makes this result unreliable.

10) There are no Y-axis titles and significant analysis in FigS1A and FigS1B.

11) The Rubsico in FigS3 is barely visible.

12) Line 316: “This was further confirmed a split luciferase” should be “…confirmed by…”.

13) Line 317: The full name of LCI assay is luciferase complementation imaging assay.

14) The authors should scrupulously check the italics in the manuscript because there are some mistakes in writing the name of genes, the overexpression lines and mutants. For example, in line 374 - line 382, the name of genes should be italicized.

15) Line 882: Rice stripe virus and Southern rice black-streaked dwarf virus should be Rice stripe virus and Southern rice black-streaked dwarf virus.

16) Line 888: “The phenotypes were observed and photos taken at 20dpi” should be “…photos were taken at 20dpi”. This mistake also occurs in many other places in the manuscript.

17) Line 902: UBQ5 should be italicized as it is a gene name. This mistake also occurs in many other places in the manuscript.

18) There is no scale bar in Fig2E.

19) The overexpression line OE-OsNF-YA2 should be italicized in Fig3A.

20) In Fig4 and FigS4, does the gene JiOsJPR10 means jasmonate inducible rice PR10 ? If so, it should be JiOsPR10 instead of JiOsJPR10.

21) Line 964: the authors claim the scale bar is 5cm in Fig4C, however, Fig4C is a statistical result, there is no scale bar at all.

22) Line 967: OsNF-YAs and OsMYCs should be italicized type.

23) Line 999: Authors claim the scale bar of the microscope photos is 50cm, it should be 50μm.

24) Fig6C – Fig6F should be marked * means p≤0.05.

Reviewer #2: The manuscript of Tan et al., explores the role of rice NF-YA family genes in antiviral defense against two different plant viruses, RSV and SRBSDV. The authors claim that OsNF-YAs physically interact with JA signaling transcription factors OsMYC2/3 and interfere with JA signaling by dissociating the OsMYC2/3-OsMED25 complex, which repress JA-mediated antiviral immunity.

Overall, the manuscript is well-written, the data generated throughout the manuscript are solid and the figure are clear. The conclusions drawn by the authors are supported by the data. This research provides valuable understanding about the interplay between NF-Y transcription factors and JA signaling in the context of plant resistance against virus in rice.

Reviewer #3: Dear Editor,

The author investigated the role of NF-Y transcription factors during viral infection with Rice stripe virus (RSV) and Southern rice black-streaked dwarf virus (SRBSDV). The author show that the virus induced overexpression of certain NF-Y genes is important for viral pathogenicity. Further experiments showed that OsNF-YAs physically interact with JA signaling transcription factors OsMYC2/3 and interfere with JA signaling by dissociating the OsMYC2/3-OsMED25 complex, which inhibits the transcriptional activation activity of OsMYC2/3. The results show that OsNF-YAs broadly inhibit plant antiviral defense by repressing JA signaling pathways. This provides new insight into how OsNF-YAs are directly associated with the JA pathway.

In general, the manuscript is well written and the conclusions are well supported by experimental evidences. The authors used different methods to support their results which makes them confident. To my opinion, these results are of high interest for the scientific community as it nicely shows how plant virus infection disturb the JA-pathway. Furthermore, it is a nice follow up of a previous study from these group.

**Part II – Major Issues: Key Experiments Required for Acceptance**

Reviewer #1: Major:

1) The authors claim that OsNF-YAs inhibit the transcriptional activation of OsMYC2/3 by direct association with the TAD domain, but the evidence is not convincing. Firstly, Fig6G - Fig6J can`t illustrate the OsNF-YA disrupt the interaction of OsMYCs and OsMED25 through TAD domain, the MYCTAD should be added to the experiment system; Secondly, BiFC and LCI assays are both imaging assays, quantitative analysis should be carried out to directly prove their interaction is disrupted. Besides, the authors should further confirm this conclusion in corresponding transgenic materials.

2) Both RSV and SRBSDV infection induce the expression of the 11 OsNF-YAs, and the authors constructed the overexpression lines of OsNF-YA1, OsNF-YA2 and OsNF-YA10, what`s the reason for choosing OE-OsNF-YA2 lines to perform RNA-seq and JA sensitivity assays?

Reviewer #2: I think this manuscript is suitable for publishing in PloS Pathogens with minor revised. The major concern is below:

They examined the effects of OsNF-YA2 on JA sensitivity by exogenous application of methyl jasmonate (MeJA). They found that overexpression of OsNF-YA2 makes rice plants less sensitive to JA treatment whereas repression of OsNF-YA2 increases sensitivity by measuring the root length. I think they should test the influence of JA on the RSV and SRBSDV resistance of OsNF-YA2 overexpressing and RNAi plants to support they conclusion that NF-YA transcription factors suppress JA signal to facilitate viral infection.

Reviewer #3: Figure 6G: Have you infiltrated a culture with an empty plasmid in the variant cYFP-OsMYC2+OsMED25-n-YFP in order to account for differences in the number of infiltrated cultures compared to the other treatments where OsNF-YA1/2? Maybe it would be good to co-infiltrate a OsNF-YA1/2 mutants unable to interact with OsMYC2/3 as control.

**Part III – Minor Issues: Editorial and Data Presentation Modifications**

Reviewer #1: Minor:

3) The authors concentrate solely on the expression level of OsNF-YAs after virus infection and the subsequent experiments, since OsNF-Y is a heterotrimeric complex consisting of OsNF-YA, OsNF-YB and OsNF-YC subunit, did OsNF-YB and OsNF-YC also play roles in antiviral defense? How do they respond to virus infection? Do they interact with OsMYC2/3?

4) The authors find OsNF-YAs negatively regulate JA pathway by qRT-PCR and JA sensitivity assays indirectly, it would be better to measure the JA contents in OE-OsNF-YA lines or Os-NF-YA RNAi mutant lines.

5) In the yeast two hybrid assay in Fig6A, MYC3N also obtain a TAD domain, it is strange that MYC3N can`t interact with OsNF-YA1 since MYC3TAD123 can interact with OsNF-YA1. Also, it would be better to label the schematic diagram in the left.

6) I don`t think the photograph of Grade II in FigS2 shows the typical symptoms of RSV infected plants.

7) In the dual-LUC reporter system in Fig6B, the authors use BD empty vector as control, they should choose a better control which can express a certain protein instead of an empty vector.

8) The authors use BiFC and LCI assays to show OsNF-YAs can imped the interaction between MYC2/3 and MED25 in Fig6.G-J. Nevertheless, they should carry out western blotting assay to exclude the effect of protein expression level.

9) Ponceau S staining in Fig1E shows the loading is not equal, which makes this result unreliable.

10) There are no Y-axis titles and significant analysis in FigS1A and FigS1B.

11) The Rubsico in FigS3 is barely visible.

12) Line 316: “This was further confirmed a split luciferase” should be “…confirmed by…”.

13) Line 317: The full name of LCI assay is luciferase complementation imaging assay.

14) The authors should scrupulously check the italics in the manuscript because there are some mistakes in writing the name of genes, the overexpression lines and mutants. For example, in line 374 - line 382, the name of genes should be italicized.

15) Line 882: Rice stripe virus and Southern rice black-streaked dwarf virus should be Rice stripe virus and Southern rice black-streaked dwarf virus.

16) Line 888: “The phenotypes were observed and photos taken at 20dpi” should be “…photos were taken at 20dpi”. This mistake also occurs in many other places in the manuscript.

17) Line 902: UBQ5 should be italicized as it is a gene name. This mistake also occurs in many other places in the manuscript.

18) There is no scale bar in Fig2E.

19) The overexpression line OE-OsNF-YA2 should be italicized in Fig3A.

20) In Fig4 and FigS4, does the gene JiOsJPR10 means jasmonate inducible rice PR10 ? If so, it should be JiOsPR10 instead of JiOsJPR10.

21) Line 964: the authors claim the scale bar is 5cm in Fig4C, however, Fig4C is a statistical result, there is no scale bar at all.

22) Line 967: OsNF-YAs and OsMYCs should be italicized type.

23) Line 999: Authors claim the scale bar of the microscope photos is 50cm, it should be 50μm.

24) Fig6C – Fig6F should be marked * means p≤0.05.

Reviewer #2: (No Response)

Reviewer #3: In principle, I suggest a minor revision with a few comments from my site:

Line 64: Wrong punctuation

Figure 1J: It must be “genes” on y-axis

Have the authors an idea whether the higher expression of OsNF-YAs is induced by specific viral protein?

How does the author explain the lower/higher viral incidence in transgenic plants? Is an effect on the vector possible?

PLOS authors have the option to publish the peer review history of their article (what does this mean?). If published, this will include your full peer review and any attached files.

Reviewer #1: No

Reviewer #2: No

Reviewer #3: **Yes: **Dr. Sebastian Liebe
---

## [Decision Letter · Decision Letter 1]

25 Apr 2022

Dear Dr. Sun,

We are pleased to inform you that your manuscript 'NF-YA transcription factors suppress jasmonic acid-mediated antiviral defense and facilitate viral infection in rice' has been provisionally accepted for publication in PLOS Pathogens.

Best regards,

Savithramma P. Dinesh-Kumar

Associate Editor

PLOS Pathogens

Peter Nagy

Section Editor

PLOS Pathogens

Kasturi Haldar

Editor-in-Chief

PLOS Pathogens

orcid.org/0000-0001-5065-158X

Michael Malim

Editor-in-Chief

PLOS Pathogens

orcid.org/0000-0002-7699-2064

Reviewer Comments (if any, and for reference):

Reviewer's Responses to Questions

**Part I - Summary**

Reviewer #1: The authors have addressed my questions and concerns. I suggest for acceptance for publication.

Reviewer #2: The authors have adderessed all my conerns and questions. It can be accepted for publication now.

Reviewer #3: (No Response)

**Part II – Major Issues: Key Experiments Required for Acceptance**

Reviewer #1: No.

Reviewer #2: (No Response)

Reviewer #3: (No Response)

**Part III – Minor Issues: Editorial and Data Presentation Modifications**

Reviewer #1: No.

Reviewer #2: (No Response)

Reviewer #3: (No Response)

PLOS authors have the option to publish the peer review history of their article (what does this mean?). If published, this will include your full peer review and any attached files.

Reviewer #1: No

Reviewer #2: No

Reviewer #3: **Yes: **Sebastian Liebe

---

## [Editor Report · Acceptance letter]

10 May 2022

Dear Dr. Sun,

We are delighted to inform you that your manuscript, "NF-YA transcription factors suppress jasmonic acid-mediated antiviral defense and facilitate viral infection in rice," has been formally accepted for publication in PLOS Pathogens.

Best regards,

Kasturi Haldar

Editor-in-Chief

PLOS Pathogens

orcid.org/0000-0001-5065-158X

Michael Malim

Editor-in-Chief

PLOS Pathogens

orcid.org/0000-0002-7699-2064